# ON THE EXPRESSIVENESS OF RATIONAL ReLU NEURAL NETWORKS WITH BOUNDED DEPTH

**Gennadiy Averkov**
BTU Cottbus-Senftenberg
averkov@b-tu.de

**Christopher Hojny**
TU Eindhoven
c.hojny@tue.nl

**Maximilian Merkert**
TU Braunschweig
m.merkert@tu-braunschweig.de

## ABSTRACT

To confirm that the expressive power of ReLU neural networks grows with their depth, the function $F_n = \max\{0, x_1, \ldots, x_n\}$ has been considered in the literature. A conjecture by Hertrich, Basu, Di Summa, and Skutella [NeurIPS 2021] states that any ReLU network that exactly represents $F_n$ has at least $\lceil \log_2(n+1) \rceil$ hidden layers. The conjecture has recently been confirmed for networks with integer weights by Haase, Hertrich, and Loho [ICLR 2023].

We follow up on this line of research and show that, within ReLU networks whose weights are decimal fractions, $F_n$ can only be represented by networks with at least $\lceil \log_3(n+1) \rceil$ hidden layers. Moreover, if all weights are $N$-ary fractions, then $F_n$ can only be represented by networks with at least $\Omega(\frac{\ln n}{\ln \ln N})$ layers. These results are a partial confirmation of the above conjecture for rational ReLU networks, and provide the first non-constant lower bound on the depth of practically relevant ReLU networks.

## 1 INTRODUCTION

An important aspect of designing neural network architectures is to understand which functions can be exactly represented by a specific architecture. Here, we say that a neural network, transforming $n$ input values into a single output value, *(exactly) represents* a function $f \colon \mathbb{R}^n \to \mathbb{R}$ if, for every input $x \in \mathbb{R}^n$, the neural network reports output $f(x)$. Understanding the expressiveness of neural network architectures can help to, among others, derive algorithms (Arora et al., 2018; Khalife et al., 2024; Hertrich & Sering, 2024) and complexity results (Goel et al., 2021; Froese et al., 2022; Bertschinger et al., 2023; Froese & Hertrich, 2023) for training networks.

One of the most popular classes of neural networks are feedforward neural networks with ReLU activation (Goodfellow et al., 2016). Their capabilities to *approximate* functions is well-studied and led to several so-called universal approximation theorems, e.g., see (Cybenko, 1989; Hornik, 1991). For example, from a result by Leshno et al. (1993) it follows that any continuous function can be approximated arbitrarily well by ReLU networks with a single hidden layer. In contrast to approximating functions, the understanding of which functions can be *exactly* represented by a neural network is much less mature. A central result by Arora et al. (2018) states that the class of functions that are exactly representable by ReLU networks is the class of continuous piecewise linear (CPWL) functions. In particular, they show that every CPWL function with $n$ inputs can be represented by a ReLU network with $\lceil \log_2(n+1) \rceil$ hidden layers. It is an open question though for which functions this number of hidden layers is also necessary.

An active research field is therefore to derive lower bounds on the number of required hidden layers. Arora et al. (2018) show that two hidden layers are necessary and sufficient to represent $\max\{0, x_1, x_2\}$ by a ReLU network. However, there is no single function which is known to require more than two hidden layers in an exact representation. In fact, Hertrich et al. (2021) formulate the following conjecture.

**Conjecture 1.** *For every integer $k$ with $1 \le k \le \lceil \log_2(n+1) \rceil$, there exists a function $f \colon \mathbb{R}^n \to \mathbb{R}$ that can be represented by a ReLU network with $k$ hidden layers, but not with $k-1$ hidden layers.*

Hertrich et al. (2021) also show that this conjecture is equivalent to the statement that any ReLU network representing $\max\{0, x_1, \ldots, x_{2^k}\}$ requires $k+1$ hidden layers. That is, if the conjecture

holds true, the lower bound of $\lceil \log_2(n+1) \rceil$ by Arora et al. (2018) is tight. While Conjecture 1 is open in general, it has been confirmed for two subclasses of ReLU networks, namely networks all of whose weights only take integer values (Haase et al., 2023) and, for $n = 4$, so-called $H$-conforming neural networks (Hertrich et al., 2021).

In this article, we follow this line of research by deriving a non-constant lower bound on the number of hidden layers in ReLU networks all of whose weights are $N$-ary fractions. Recall that a rational number is an $N$-ary fraction if it can be written as $\frac{z}{N^t}$ for some integer $z$ and non-negative integer $t$.

**Theorem 2.** *Let $n$ and $N$ be positive integers, and let $p$ be a prime number that does not divide $N$. Every ReLU network with weights being $N$-ary fractions requires at least $\lceil \log_p(n+1) \rceil$ hidden layers to exactly represent the function $\max\{0, x_1, \ldots, x_n\}$.*

**Corollary 3.** *Every ReLU network all of whose weights are decimal fractions requires at least $\lceil \log_3(n+1) \rceil$ hidden layers to exactly represent $\max\{0, x_1, \ldots, x_n\}$.*

While Theorem 2 does not resolve Conjecture 1 because it makes no statement about general real weights, note that in most applications floating point arithmetic is used (IEEE, 2019). That is, in neural network architectures used in practice, one is actually restricted to weights being $N$-ary fractions. Moreover, when quantization, see, e.g., (Gholami et al., 2022) is used to make neural networks more efficient in terms of memory and speed, weights can become low-precision decimal numbers, cf., e.g., (Nagel et al., 2020). Consequently, Theorem 2 provides, to the best of our knowledge, the first non-constant lower bound on the depth of practically relevant ReLU networks.

Relying on Theorem 2, we also derive the following lower bound.

**Theorem 4.** *There is a constant $C > 0$ such that, for all integers $n, N \geq 3$, every ReLU network with weights being $N$-ary fractions that represents $\max\{0, x_1, \ldots, x_n\}$ has depth at least $C \cdot \frac{\ln n}{\ln \ln N}$.*

Theorem 4, in particular, shows that there is no constant-depth ReLU network that exactly represents $\max\{0, x_1, \ldots, x_n\}$ if all weights are rational numbers all having a common denominator $N$.

In view of the integral networks considered by Haase et al. (2023), we stress that our results do not simply follow by scaling integer weights to rationals, which has already been discussed in Haase et al. (2023, Sec. 1.3). We therefore extend the techniques by Haase et al. (2023) to make use of number theory and polyhedral combinatorics to prove our results that cover standard number representations of rationals on a computer.

**Outline** To prove our main results, Theorems 2 and 4, the rest of the paper is structured as follows. First, we provide some basic definitions regarding neural networks that we use throughout the article, and we provide a brief overview of related literature. Section 2 then provides a short summary of our overall strategy to prove Theorems 2 and 4 as well as some basic notation. The different concepts of polyhedral theory and volumes needed in our proof strategy are detailed in Section 2.1, whereas Section 2.2 recalls a characterization of functions representable by a ReLU neural network from the literature, which forms the basis of our proofs. In Section 3, we derive various properties of polytopes associated with functions representable by a ReLU neural network, which ultimately allows us to prove our main results in Section 3.3. The paper is concluded in Section 4.

**Basic Notation for ReLU Networks** To describe the neural networks considered in this article, we introduce some notation. We denote by $\mathbb{Z}$, $\mathbb{N}$, and $\mathbb{R}$ the sets of integer, positive integer, and real numbers, respectively. Moreover, $\mathbb{Z}_+$ and $\mathbb{R}_+$ denote the sets of non-negative integers and reals, respectively.

Let $k \in \mathbb{Z}_+$. A *feedforward neural network with rectified linear units (ReLU)* (or simply *ReLU network* in the following) with $k + 1$ layers can be described by $k + 1$ affine transformations $t^{(1)} \colon \mathbb{R}^{n_0} \to \mathbb{R}^{n_1}, \ldots, t^{(k+1)} \colon \mathbb{R}^{n_k} \to \mathbb{R}^{n_{k+1}}$. It *exactly represents* a function $f \colon \mathbb{R}^n \to \mathbb{R}$ if and only if $n_0 = n$, $n_{k+1} = 1$, and the alternating composition

$$t^{(k+1)} \circ \sigma \circ t^{(k)} \circ \sigma \circ \cdots \circ t^{(2)} \circ \sigma \circ t^{(1)}$$

coincides with $f$, where, by slightly overloading notation, $\sigma$ denotes the component-wise application of the *ReLU activation function* $\sigma \colon \mathbb{R} \to \mathbb{R}$, $\sigma(x) = \max\{0, x\}$ to vectors in any dimension. For each $i \in \{1, \ldots, k+1\}$ and $x \in \mathbb{R}^{n_{i-1}}$, let $t^{(i)}(x) = A^{(i)}x + b^{(i)}$ for some $A^{(i)} \in \mathbb{R}^{n_i \times n_{i-1}}$ and $b^{(i)} \in \mathbb{R}^{n_i}$. The entries of $A^{(i)}$ are called *weights* and those of $b^{(i)}$ are called *biases* of the network. The network's *depth* is $k + 1$, and the *number of hidden layers* is $k$.

The set of all functions $\mathbb{R}^n \to \mathbb{R}$ that can be represented exactly by a ReLU network of depth $k+1$ is denoted by $\mathrm{ReLU}_n(k)$. Moreover, if $R \subseteq \mathbb{R}$ is a ring, we denote by $\mathrm{ReLU}_n^R(k)$ the set of all functions $\mathbb{R}^n \to \mathbb{R}$ that can be represented exactly by a ReLU network of depth $k+1$ all of whose weights are contained in $R$. Throughout this paper, we will mainly work with the rings $\mathbb{Z}$, $\mathbb{R}$, or the ring of $N$-ary fractions.

The set $\mathrm{ReLU}_n^R(0)$ is the set of affine functions $f(x_1, \ldots, x_n) = b + a_1 x_1 + \cdots + a_n x_n$ with $b \in \mathbb{R}$, and $a_1, \ldots, a_n \in R$. It can be directly seen from the definition of ReLU networks that, for $k \in \mathbb{N}$, one has $f \in \mathrm{ReLU}_n^R(k)$ if and only if $f(x) = u_0 + u_1 \max\{0, g_1(x)\} + \cdots + u_m \max\{0, g_m(x)\}$ holds for some $m \in \mathbb{N}$, $u_0 \in \mathbb{R}$, $u_1, \ldots, u_m \in R$, and functions $g_1, \ldots, g_m \in \mathrm{ReLU}_n^R(k-1)$.

**Related Literature**  Regarding the expressiveness of ReLU networks, Hertrich et al. (2021) show that four layers are needed to exactly represent $\max\{0, x_1, \ldots, x_4\}$ if the network satisfies the technical condition of being $H$-conforming. By restricting the weights of a ReLU network to be integer, Haase et al. (2023) prove that $\mathrm{ReLU}_n^{\mathbb{Z}}(k-1) \subsetneq \mathrm{ReLU}_n^{\mathbb{Z}}(k)$ for every $k \leq \lceil \log_2(n+1) \rceil$. In particular, $\max\{0, x_1, \ldots, x_{2^k}\} \notin \mathrm{ReLU}_{2^k}^{\mathbb{Z}}(k)$. If the activation function is changed from ReLU to $x \mapsto \mathbb{1}_{\{x>0\}}$, Khalife et al. (2024) show that two hidden layers are both necessary and sufficient for all functions representable by such a network.

If one is only interested in approximating a function, Safran et al. (2024) show that $\max\{0, x_1, \ldots, x_n\}$ can be approximated arbitrarily well by $\mathrm{ReLU}_n^{\mathbb{Z}}(2)$-networks of width $n(n+1)$ with respect to the $L_2$ norm for continuous distributions. By increasing the depth of these networks, they also derive upper bounds on the required width in such an approximation. The results by Safran et al. (2024) belong to the class of so-called universal approximation theorems, which describe the ability to approximate classes of functions by specific types of neural networks, see, e.g., (Cybenko, 1989; Hornik, 1991; Barron, 1993; Pinkus, 1999; Kidger & Lyons, 2020). However, Vardi & Shamir (2020) show that there are significant theoretical barriers for depth-separation results for polynomially-sized $\mathrm{ReLU}_n(k)$-networks for $k \geq 3$, by establishing links to the separation of threshold circuits as well as to so-called natural-proof barriers. When taking specific data into account, Lee et al. (2024) derive lower and upper bounds on both the depth and width of a neural network that correctly classifies a given data set. More general investigations of the relation between the width and depth of a neural network are discussed, among others, by Arora et al. (2018); Eldan & Shamir (2016); Hanin (2019); Raghu et al. (2017); Safran & Shamir (2017); Telgarsky (2016).

## 2  PROOF STRATEGY AND THEORETICAL CONCEPTS

To prove Theorems 2 and 4, we extend the ideas of Haase et al. (2023). We therefore provide a very concise summary of the arguments of Haase et al. (2023). Afterwards, we briefly mention the main ingredients needed in our proofs, which are detailed in the following subsections.

A central ingredient for the results by Haase et al. (2023) is a polyhedral characterization of all functions in $\mathrm{ReLU}_n(k)$, which has been derived by Hertrich (2022). This characterization links functions representable by a ReLU network and so-called support functions of polytopes $P \subseteq \mathbb{R}^n$ all of whose vertices belong to $\mathbb{Z}^n$, so-called *lattice polytopes*. It turns out that the function $\max\{0, x_1, \ldots, x_n\}$ in Theorems 2 and 4 can be expressed as the support function of a particular lattice polytope $P_n \subseteq \mathbb{R}^n$. By using a suitably scaled version $\mathrm{Vol}_n$ of the classical Euclidean volume in $\mathbb{R}^n$, one can achieve $\mathrm{Vol}_n(P) \in \mathbb{Z}$ for all lattice polytopes $P \subseteq \mathbb{R}^n$. Haase et al. (2023) then show that, if the support function $h_P$ of a lattice polytope $P \subseteq \mathbb{R}^n$ can be exactly represented by a ReLU network with $k$ hidden layers, all faces of $P$ of dimension at least $2^k$ have an even normalized volume. For $n = 2^k$, however, $\mathrm{Vol}_n(P_n)$ is odd. Hence, its support function cannot be represented by a ReLU network with $k$ hidden layers.

We show that the arguments of Haase et al. (2023) can be adapted by replacing the divisor 2 with an arbitrary prime number $p$. Another crucial insight is that the theory of mixed volumes can be used to analyze the behavior of $\mathrm{Vol}_n(A+B)$ for the Minkowski sum $A + B \coloneqq \{a+b \colon a \in A, b \in B\}$ of lattice polytopes $A, B \subset \mathbb{R}^n$. The Minkowski-sum operation is also involved in the polyhedral characterization of Hertrich (2022), and so it is also used by Haase et al. (2023), who provide a version of Theorem 2 for integer weights. They, however, do not directly use mixed volumes. A key observation used in our proofs, and obtained by a direct application of mixed volumes, is that the

map associating to a lattice polytope $P$ the coset of $\mathrm{Vol}_n(P)$ modulo a prime number $p$ is additive when $n$ is a power of $p$. Combining these ingredients yields Theorems 2 and 4.

**Some Basic Notation** The standard basis vectors in $\mathbb{R}^n$ are denoted by $e_1, \ldots, e_n$, whereas $0$ denotes the null vector in $\mathbb{R}^n$. Throughout the article, all vectors $x \in \mathbb{R}^n$ are column vectors, and we denote the transposed vector by $x^\top$. If $x$ is contained in the integer lattice $\mathbb{Z}^n$, we call it a *lattice point*. For vectors $x, y \in \mathbb{R}^n$, their scalar product is given by $x^\top y$. For $m \in \mathbb{N}$, we will write $[m]$ for the set $\{1, \ldots, m\}$. The convex-hull operator is denoted by $\mathrm{conv}$, and the base-$b$ logarithm by $\log_b$, while the natural logarithm is denoted $\ln$.

The central function of this article is $\max\{0, x_1, \ldots, x_n\}$, which we abbreviate by $F_n$.

## 2.1 Basic Properties of Polytopes and Lattice Polytopes

As outlined above, the main tools needed to prove Theorems 2 and 4 are polyhedral theory and different concepts of volumes. This section summarizes the main concepts and properties that we need in our argumentation in Section 3. For more background, we refer the reader to the monographs (Beck & Robins, 2020; Hug & Weil, 2020; Schneider, 2014).

**Polyhedra, Lattice Polyhedra, and Their Normalized Volume** A *polytope* $P \subseteq \mathbb{R}^n$ is the convex hull $\mathrm{conv}(p_1, \ldots, p_m)$ of finitely many points $p_1, \ldots, p_m \in \mathbb{R}^n$. We introduce the family

$$\mathcal{P}(S) \coloneqq \{\mathrm{conv}(p_1, \ldots, p_m)\colon m \in \mathbb{N},\ p_1, \ldots, p_m \in S\}$$

of all non-empty polytopes with vertices in $S \subseteq \mathbb{R}^n$. Thus, $\mathcal{P}(\mathbb{R}^n)$ is the family of all polytopes in $\mathbb{R}^n$ and $\mathcal{P}(\mathbb{Z}^n)$ is the family of all *lattice polytopes* in $\mathbb{R}^n$. For $d \in \{0, \ldots, n\}$, we also introduce the family

$$\mathcal{P}_d(S) \coloneqq \{P \in \mathcal{P}(S)\colon \dim(P) \leq d\}.$$

of polytopes of dimension at most $d$, where the dimension of a polytope $P$ is defined as the dimension of its affine hull, i.e., the smallest affine subspace of $\mathbb{R}^n$ containing $P$. The *Euclidean volume* $\mathrm{vol}_n$ on $\mathbb{R}^n$ is the $n$-dimensional Lebesgue measure, scaled so that $\mathrm{vol}_n$ is equal to 1 on the unit cube $[0, 1]^d$. Note that measure-theoretic subtleties play no role in our context since we restrict the use of $\mathrm{vol}_n$ to $\mathcal{P}(\mathbb{R}^n)$. The *normalized volume* $\mathrm{Vol}_n$ in $\mathbb{R}^n$ is the $n$-dimensional Lebesgue measure normalized so that $\mathrm{Vol}_n$ is equal to 1 on the *standard simplex* $\Delta_n \coloneqq \mathrm{conv}(0, e_1, \ldots, e_n)$. Clearly, $\mathrm{Vol}_n = n! \cdot \mathrm{vol}_n$ and $\mathrm{Vol}_n$ takes non-negative integer values on lattice polytopes.

**Support Functions** For a polytope $P = \mathrm{conv}(p_1, \ldots, p_m) \subseteq \mathbb{R}^n$, its *support function* is

$$h_P(x) \coloneqq \max\{x^\top y : y \in P\},$$

and it is well-known that $h_P(x) = \max\{p_1^\top x, \ldots, p_m^\top x\}$. Consequently, $\max\{0, x_1, \ldots, x_n\}$ from Theorems 2 and 4 is the support function of $\Delta_n$.

**Mixed Volumes** For sets $A, B \subseteq \mathbb{R}^n$, we introduce the *Minkowski sum*

$$A + B \coloneqq \{a + b\colon a \in A, b \in B\}$$

and the multiplication

$$\lambda A = \{\lambda a\colon a \in A\}$$

of $A$ by a non-negative factor $\lambda \in \mathbb{R}_+$. For an illustration of the Minkowski sum, we refer to Figure 2. Note that, if $S \in \{\mathbb{R}^n, \mathbb{Z}^n\}$ and $A, B \in \mathcal{P}(S)$, then $A + B \in \mathcal{P}(S)$, too. If $A$ and $B$ are (lattice) polytopes, then $A + B$ is also a (lattice) polytope, and the support functions of $A, B$ and $A + B$ are related by $h_{A+B} = h_A + h_B$.

If $(G, +)$ is an Abelian semi-group (i.e., a set with an associative and commutative binary operation), we call a map $\phi\colon \mathcal{P}(\mathbb{R}^n) \to G$ *Minkowski additive* if the Minkowski addition on $\mathcal{P}(\mathbb{R}^n)$ gets preserved by $\phi$ in the sense that $\phi(A + B) = \phi(A) + \phi(B)$ holds for all $A, B \in \mathcal{P}(\mathbb{R}^n)$.

The following is a classical result of Minkowski.

**Theorem 5** (see, e.g., (Schneider, 2014, Ch. 5))**.** *There exists a unique functional, called the* mixed volume*,*

$$\mathrm{V}\colon \mathcal{P}(\mathbb{R}^n)^n \to \mathbb{R},$$

*with the following properties valid for all $P_1, \ldots, P_n, A, B \in \mathcal{P}(\mathbb{R}^n)$ and $\alpha, \beta \in \mathbb{R}_+$:*

*(a)* V *is invariant under permutations, i.e.* $\mathrm{V}(P_1, \ldots, P_n) = \mathrm{V}(P_{\sigma(1)}, \ldots, P_{\sigma(n)})$ *for every permutation $\sigma$ on $[n]$.*

*(b)* V *is Minkowski linear in all input parameters, i.e., for all $i \in [n]$, it holds that*

$$\begin{aligned} \mathrm{V}(P_1, \ldots P_{i-1}, \alpha A + \beta B, P_{i+1}, \ldots, P_n) =& \alpha \mathrm{V}(P_1, \ldots P_{i-1}, A, P_{i+1}, \ldots, P_n) \\ &+ \beta \mathrm{V}(P_1, \ldots P_{i-1}, B, P_{i+1}, \ldots, P_n) \end{aligned}$$

*(c)* V *is equal to $\mathrm{Vol}_n$ on the diagonal, i.e.,* $\mathrm{V}(A, \ldots, A) = \mathrm{Vol}_n(A)$.

We refer to Chapter 5 of the monograph by Schneider (2014) on the Brunn-Minkowski theory for more information on mixed volumes, where also an explicit formula for the mixed volume is presented. Our definition of the mixed volume differs by a factor of $n!$ from the definition in Schneider (2014) since we use the normalized volume $\mathrm{Vol}_n$ rather than the Euclidean volume $\mathrm{vol}_n$ to fix $\mathrm{V}(P_1, \ldots, P_n)$ in the case $P_1 = \ldots = P_n$. Our way of introducing mixed volumes is customary in the context of algebraic geometry. It is known that, for this normalization, $\mathrm{V}(P_1, \ldots, P_n) \in \mathbb{Z}_+$ when $P_1, \ldots, P_n$ are lattice polytopes; see, for example, (Maclagan & Sturmfels, 2015, Ch. 4.6). From the defining properties one can immediately see that, for $A, B \in \mathcal{P}(\mathbb{R}^n)$, one has the analogue of the binomial formula, which we will prove in Appendix A.2 for the sake of completeness:

$$\mathrm{Vol}_n(A + B) = \sum_{i=0}^{n} \binom{n}{i} \mathrm{V}(\underbrace{A, \ldots, A}_{i}, \underbrace{B, \ldots, B}_{n-i}). \tag{1}$$

**Normalized Volume of Non-Full-Dimensional Polytopes** So far, we have introduced the normalized volume $\mathrm{Vol}_n \colon \mathcal{P}(\mathbb{R}^n) \to \mathbb{R}_+$, i.e., if $P \in \mathcal{P}(\mathbb{R}^n)$ is not full-dimensional, then $\mathrm{Vol}_n(P) = 0$. We also associate with a polytope $P \in \mathcal{P}_d(\mathbb{Z}^n)$ of dimension at most $d$ an appropriately normalized $d$-dimensional volume by extending the use of $\mathrm{Vol}_d \colon \mathcal{P}(\mathbb{Z}^d) \to \mathbb{Z}_+$ to $\mathrm{Vol}_d \colon \mathcal{P}_d(\mathbb{Z}^n) \to \mathbb{Z}_+$. In the case $\dim(P) < d$, we define $\mathrm{Vol}_d(P) = 0$. If $d = 0$, let $\mathrm{Vol}_d(P) = 1$. In the non-degenerate case $d = \dim(P) \in \mathbb{N}$, we fix $Y$ to be the affine hull of $P$ and consider a bijective affine map $T \colon \mathbb{R}^d \to Y$ satisfying $T(\mathbb{Z}^d) = Y \cap \mathbb{Z}^n$. For such choice of $T$, we have $T^{-1}(P) \in \mathcal{P}(\mathbb{Z}^d)$. It turns out that the $d$-dimensional volume of $T^{-1}(P)$ depends only on $P$ and not on $T$ so that we define $\mathrm{Vol}_d(P) := \mathrm{Vol}_d(T^{-1}(P))$. Based on (Beck & Robins, 2020, Corollary 3.17 and §5.4), there is the following intrinsic way of introducing $\mathrm{Vol}_d(P)$. Let $G(P)$ denote the number of lattice points in $P$. Then, for $t \in \mathbb{Z}_+$, one has $\mathrm{Vol}_d(P) := d! \cdot \lim_{t \to \infty} \frac{1}{t^d} G(tP)$.

**Remark 6.** *For every $d$-dimensional affine subspace $Y \subseteq \mathbb{R}^n$ which is affinely spanned by $d + 1$ lattice points, we can define $\mathrm{Vol}_d$ for every polytope $P \in \mathcal{P}(Y)$, which is not necessarily a lattice polytope, by the same formula $\mathrm{Vol}_d(P) := \mathrm{Vol}_d(T^{-1}(P))$, using an auxiliary map $T \colon \mathbb{R}^d \to Y$ described above. Consequently, by replacing the dimension $n$ with $d$ and the family of polytopes $\mathcal{P}(\mathbb{R}^n)$ with the family $\mathcal{P}(Y)$ in Minkowski's Theorem 5, we can introduce the notion of mixed volumes for polytopes in $\mathcal{P}(Y)$. More specifically, we will make use of the mixed volumes of lattice polytopes in $\mathcal{P}(Y \cap \mathbb{Z}^n)$.*

**Normalized Volume of the Affine Join** The following proposition, borrowed from Haase et al. (2023), addresses the divisibility properties of the convex hull of the union of lattice polytopes that lie in skew affine subspaces.

**Proposition 7** (Haase et al. 2023, Lemma 6). *Let $A, B \in \mathcal{P}(\mathbb{Z}^n)$ be polytopes of dimensions $i \in \mathbb{Z}_+$ and $j \in \mathbb{Z}_+$, respectively, such that $P := \mathrm{conv}(A \cup B)$ is of dimension $i + j + 1$. Then $\mathrm{Vol}_{i+j}(P)$ is divisible by $\mathrm{Vol}_i(A) \mathrm{Vol}_j(B)$. In particular, if $i = 0$, then $P$ is a pyramid over $B$ whose normalized volume $\mathrm{Vol}_{1+j}(B)$ is divisible by the normalized volume $\mathrm{Vol}_j(B)$ of its base $B$.*

For an example illustration, see Figure 1. Since $P_1$ and $P_2$ lie in skew affine subspaces, Proposition 7 applies. Indeed, $\mathrm{Vol}_3(\mathrm{conv}(P_1 \cup P_2)) = 12$ is divisible by $\mathrm{Vol}_2(P_1) = 6$ (and $\mathrm{Vol}_0(P_2) = 1$).

### 2.2 A POLYHEDRAL CRITERION FOR FUNCTIONS REPRESENTABLE WITH $k$ HIDDEN LAYERS

Next to the geometric concepts that we discussed before, the second main building block of our proofs is the polyhedral characterization of $\mathrm{ReLU}_n(k)$ by Hertrich (2022). In the following, we introduce the necessary concepts and present Hertrich's characterization.

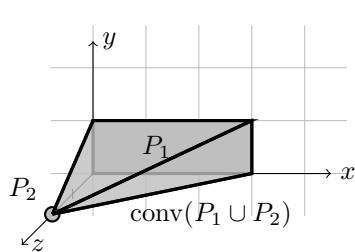

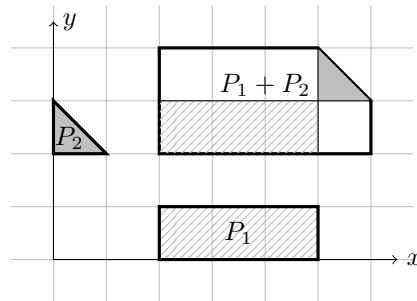

Figure 1: Illustration of the convex hull of a polytope and a point, relating to Proposition 7.

Figure 2: Illustration of the Minkowski sum of two polytopes, relating to Example 12.

Note that $F_n$ is *positively homogeneous*, i.e., for all $\lambda \in \mathbb{R}_+$ and $x \in \mathbb{R}^n$, one has $F_n(\lambda x) = \lambda F_n(x)$. For positively homogeneous functions $f$, Hertrich et al. (2021) show that $f \in \mathrm{ReLU}_n(k)$ if and only if there exists a ReLU network of depth $k+1$ all of whose biases are 0. This result easily generalizes to ReLU networks with weights being restricted to a ring $R$. We therefore denote by $\mathrm{ReLU}_n^{R,0}(k)$ the set of all $n$-variate positively homogeneous functions representable by ReLU networks with $k$ hidden layers, weights in $R$, and all biases being 0. Moreover, $\mathrm{ReLU}_n^{R,0} := \bigcup_{k=0}^{\infty} \mathrm{ReLU}_n^{R,0}(k)$.

To formulate the characterization by Hertrich (2022), we define the *sum-union closure* for a family of polytopes $\mathcal{X}$ in $\mathbb{R}^n$ as

$$\mathrm{SU}(\mathcal{X}) := \left\{ \sum_{i=1}^{m} \mathrm{conv}(A_i \cup B_i) \colon m \in \mathbb{N},\ A_i, B_i \in \mathcal{X}, i \in [m] \right\}.$$

The $k$-fold application of the operation gives the *$k$-fold sum-union closure* $\mathrm{SU}^k(\mathcal{X})$. In other words, $\mathrm{SU}^0(\mathcal{X}) = \mathcal{X}$ and $\mathrm{SU}^k(\mathcal{X}) = \mathrm{SU}(\mathrm{SU}^{k-1}(\mathcal{X}))$ for $k \in \mathbb{N}$. We will apply the $k$-fold sum-union closure to $\mathcal{P}_0(S)$, the set of all 0-dimensional polytopes of the form $\{s\}$, with $s \in S$.

The set $\mathrm{SU}^k(\mathcal{X})$ forms a semi-group with respect to Minkowski-addition since, directly from the representation of elements of $\mathrm{SU}^k(\mathcal{X})$ as sums with arbitrarily many summands, one sees that $\mathrm{SU}^k(\mathcal{X})$ is closed under Minkowski addition.

**Theorem 8** ((Hertrich, 2022, Thm. 3.35) for $R = \mathbb{R}$ and (Haase et al., 2023, Thm. 8) for $R = \mathbb{Z}$). *Let $R$ be $\mathbb{R}$ or $\mathbb{Z}$. Then*
$$\mathrm{ReLU}_n^{R,0}(k) = \{h_A - h_B \colon A, B \in \mathrm{SU}^k(\mathcal{P}_0(R^n))\}.$$

**Corollary 9.** *Let $k \in \mathbb{Z}_+$ and $R$ be $\mathbb{R}$ or $\mathbb{Z}$. Let $P \in \mathcal{P}(R^n)$. Then, the support function $h_P$ of $P$ satisfies $h_P \in \mathrm{ReLU}_n^R(k)$ if and only if $P + A = B$ for some $A, B \in \mathrm{SU}^k(\mathcal{P}_0(R^n))$.*

*Proof.* By Theorem 8, we have that $h_P \in \mathrm{ReLU}_n^R(k)$ if and only if $h_P = h_B - h_A$ for some $A, B \in \mathrm{SU}^k(\mathcal{P}_0(R^n))$. The equation $h_P = h_B - h_A$ can be rewritten as $h_B = h_P + h_A = h_{P+A}$, as support functions are Minkowski additive. Furthermore, every polytope is uniquely determined by its support function, see (Hug & Weil, 2020), so $h_{P+A} = h_B$ is equivalent to $P + A = B$. $\qquad\square$

The characterization of $\mathrm{ReLU}_n^{R,0}(k)$ via $\mathrm{SU}^k(\mathcal{P}_0(R^n))$ as well as the geometric concepts of volumes will allow us to prove Theorem 2. The core step of our proof is to show that $F_n$, which is the support function of $\Delta_n$, is not contained in $\mathrm{ReLU}_n^{\mathbb{Z},0}(k)$ for small $k$. As we will see later, it turns out to be useful to not work exclusively with full-dimensional polytopes in $\mathrm{SU}^k(\mathcal{P}_0(\mathbb{Z}^n))$, but with some of their lower-dimensional faces. The next lemma provides the formal mechanism that we use, namely if $P \in \mathrm{SU}^k(\mathcal{P}_0(\mathbb{Z}^n))$ and $F$ is a face of $P$, then $h_F \notin \mathrm{ReLU}_n^{\mathbb{Z}}(k)$ implies also $h_P \notin \mathrm{ReLU}_n^{\mathbb{Z}}(k)$. We defer the lemma's proof to Appendix A.1.1.

**Lemma 10.** *Let $k \in \mathbb{Z}_+$. Then, for all $P \in \mathrm{SU}^k(\mathcal{P}_0(\mathbb{Z}^n))$ and $u \in \mathbb{R}^n$, the face of $P$ in direction $u$, given by*
$$P^u := \{x \in P \colon u^\top x = h_P(u)\},$$
*belongs to $\mathrm{SU}^k(\mathcal{P}_0(\mathbb{Z}^n))$. In other words, $\mathrm{SU}^k(\mathcal{P}_0(\mathbb{Z}^n))$ is closed under taking non-empty faces.*

## 3   RESULTS AND PROOFS

The goal of this section is to prove Theorems 2 and 4 for the ring $R$ of $N$-ary fractions. To this end, we will rescale $F_n$ by a suitable scalar $\lambda \in \mathbb{N}$ such that the containment $F_n \in \mathrm{ReLU}_n^R(k)$ is equivalent to $\lambda F_n \in \mathrm{ReLU}_n^{\mathbb{Z}}(k)$. To show that $\lambda F_n \notin \mathrm{ReLU}_n^{\mathbb{Z}}(k)$ if $k$ is too small, we use a volume-based argument. More precisely, we show that, for lattice polytopes $P \subseteq \mathbb{R}^n$ whose support functions $h_P$ are contained in $\mathrm{ReLU}_n^{\mathbb{Z}}(k)$ and suitably defined dimensions $d$ and prime numbers $p$, their volumes $\mathrm{Vol}_d(P)$ are divisible by $p$. In contrast, $\mathrm{Vol}_d(\lambda \Delta_n)$ is not divisible by $p$, and thus, $\lambda F_n \notin \mathrm{ReLU}_n^{\mathbb{Z}}(k)$. This strategy is inspired by the proof of Haase et al. (2023) for $F_n \notin \mathrm{ReLU}_n^{\mathbb{Z}}(k)$, where related results are shown for the special case $p = 2$. Our results, however, are more general and do not follow directly from their results.

To pursue this strategy, Sections 3.1 and 3.2 derive novel insights into volumes $\mathrm{Vol}_d(P)$ of lattice polytopes $P$ whose support functions $h_p$ are contained in $\mathrm{ReLU}_n^{\mathbb{Z}}(k)$. These insights are then used in Section 3.3 to prove Theorems 2 and 4.

### 3.1   DIVISIBILITY OF NORMALIZED VOLUMES BY A PRIME

To understand the divisibility of $\mathrm{Vol}_d$ by a prime number mentioned above, we investigate cases in which $\mathrm{Vol}_d \colon \mathcal{P}_d(\mathbb{Z}^n) \to \mathbb{Z}$ modulo a prime is Minkowski additive. To make this precise, we introduce some notation.

For $a, b \in \mathbb{Z}$ and $m \in \mathbb{N}$ we write $a \equiv_m b$ if $a - b$ is divisible by $m$. This is called the congruence of $a$ and $b$ modulo $m$. The coset $[z]_m$ of $z \in \mathbb{Z}$ modulo $m$ is the set of all integers congruent to $z$ modulo $m$, and we denote the set of all such cosets by $\mathbb{Z}_m$. The addition of cosets is defined by $[a]_m + [b]_m := [a+b]_m$ for $a, b \in \mathbb{Z}$. Endowing $\mathbb{Z}_m$ with the addition operation $+$ yields a group of order $m$.

The following is an easy-to-prove but crucial observation. It states that when we consider lattice polytopes in a $d$-dimenensional subspace $Y \subseteq \mathbb{R}^n$ spanned by $d$ lattice points, the volume $\mathrm{Vol}_d$, taken modulo a prime number $p$, is an additive functional when $d$ is a power of $p$.

**Proposition 11.** *Let $d = p^t \leq n$ be a power of a prime number $p$, with $t \in \mathbb{N}$. Let $P_1, \ldots, P_m \in \mathcal{P}_d(\mathbb{Z}^n)$ be such that $\sum_{i=1}^m P_i \in \mathcal{P}_d(\mathbb{Z}^n)$. Then,*

$$\mathrm{Vol}_d\left(\sum_{i=1}^m P_i\right) \equiv_p \sum_{i=1}^m \mathrm{Vol}_d(P_i).$$

*Proof.* Note that by the assumption $\sum_{i=1}^m P_i \in \mathcal{P}_d(\mathbb{Z}^d)$ all of the $P_i$'s lie, up to appropriate translation, in a $d$-dimensional vector subspace $Y$ of $\mathbb{R}^d$, which is spanned by $d$ lattice points. There is no loss of generality in assuming that $P_i \subseteq Y$ and, in view of Remark 6, we can use the mixed volume functional on $d$-tuples of polytopes from $\mathcal{P}(Y)$, which will give an integer value for polytopes in $\mathcal{P}(Y \cap \mathbb{Z}^n)$. By an inductive argument, it is sufficient to consider the case $m = 2$. It is well known that if $d$ is a power of $p$, the binomial coefficients $\binom{d}{1}, \ldots, \binom{d}{d-1}$ in (1) are divisible by $p$, see, e.g., Mihet (2010, Cor. 2.9). Thus, (1) implies $\mathrm{Vol}_d(P_1 + P_2) \equiv_p \mathrm{Vol}_d(P_1) + \mathrm{Vol}_d(P_2)$ for $P_1, P_2 \in \mathcal{P}(Y \cap \mathbb{Z}^n)$. $\square$

**Example 12.** Consider the polytope $P_1 + P_2 \in \mathcal{P}_2(\mathbb{Z}^2)$ for the rectangle $P_1 = [2,5] \times [0,1] \in \mathcal{P}_2(\mathbb{Z}^2)$ and the shifted standard simplex $P_2 = \Delta_2 + \{(0,2)^\top\} \in \mathcal{P}_2(\mathbb{Z}^2)$ as depicted in Figure 2. In the picture, $P_1 + P_2$ is decomposed into regions in such a way that the volume of the mixed area $\mathrm{V}(P_1, P_2)$ can be read off. In view of the equality $\mathrm{Vol}_2(P_1 + P_2) = \mathrm{V}(P_1 + P_2, P_1 + P_2) = \mathrm{V}(P_1, P_1) + 2\,\mathrm{V}(P_1, P_2) + \mathrm{V}(P_2, P_2) = \mathrm{Vol}_2(P_1) + 2\,\mathrm{V}(P_1, P_2) + \mathrm{Vol}_2(P_2)$, see (1), the total volume of the unshaded part of $P_1 + P_2$ must be exactly $2\,\mathrm{V}(P_1, P_2)$. For $p = 2$ we have $\mathrm{Vol}_2(P_1 + P_2) = 15 \equiv_2 6 + 1 = \mathrm{Vol}_2(P_1) + \mathrm{Vol}_2(P_2)$, i.e., the parity of $\mathrm{Vol}_2(P_1 + P_2)$ is indeed that of $\mathrm{Vol}_2(P_1) + \mathrm{Vol}_2(P_2)$. In contrast, divisibility by $p = 3$ does not match, as $15 \not\equiv_3 7$. However, this does not contradict Proposition 11, as $d = 2$ is not a power of $p = 3$.

To derive divisibility properties of $\mathrm{Vol}_d(P)$ for lattice polytopes $P$ with $h_P \in \mathrm{ReLU}_n^{\mathbb{Z}}(k)$, we make use of the characterization of $\mathrm{ReLU}_n^{\mathbb{Z}}(k)$ via the SU-operator. Recall that one of the two defining

operations of SU is $\mathrm{conv}(A \cup B)$ for suitable polytopes $A$ and $B$. A crucial observation is that for certain dimensions $d$, the divisibility of $\mathrm{Vol}_d(\mathrm{conv}(A \cup B))$ by a prime number is inherited from particular lower-dimensional faces of $A$ and $B$.

**Proposition 13.** *Let $d = p^t \leq n$ be a power of a prime number $p$, with $t \in \mathbb{N}$. Moreover, let $P = \mathrm{conv}(A \cup B) \in \mathcal{P}_d(\mathbb{Z}^n)$ for $A, B \in \mathcal{P}_d(\mathbb{Z}^n)$. If $\mathrm{Vol}_{p^{t-1}}(F) \equiv_p 0$ for all $p^{t-1}$-dimensional faces $F$ of $A$ and $B$, then $\mathrm{Vol}_{p^t}(P) \equiv_p 0$.*

Note that this result also holds trivially if no face of dimension $p^{t-1}$ exists. We defer the proof of this result to Appendix A.1.2.

## 3.2 MODULAR OBSTRUCTION ON VOLUME FOR REALIZABILITY WITH $k$ HIDDEN LAYERS

Equipped with the previously derived results, we have all ingredients together to prove the aforementioned results on the divisibility of $\mathrm{Vol}_d(P)$ for lattice polytopes $P$ with $h_P \in \mathrm{ReLU}_n^{\mathbb{Z}}(k)$.

**Theorem 14.** *Let $d = p^t \leq n$ be a power of a prime number $p$, with $t \in \mathbb{N}$. Let $k \in [t]$ and $P \in \mathrm{SU}^k(\mathcal{P}_0(\mathbb{Z}^n))$. Then $\mathrm{Vol}_{p^k}(F) \equiv_p 0$ for all $p^k$-dimensional faces $F$ of $P$.*

*Proof.* We argue by induction on $k$. If $k = 1$, then $\mathrm{SU}^1(\mathcal{P}_0(\mathbb{Z}^n))$ consists of lattice zonotopes. These are polytopes of the form $P = S_1 + \cdots + S_m$, where $S_1, \ldots, S_m$ are line segments joining a pair of lattice points. One has $\mathrm{Vol}_d(P) \equiv_p \mathrm{Vol}_d(\sum_{i=1}^m S_i) \equiv_p \sum_{i=1}^m \mathrm{Vol}_d(S_i)$, by Proposition 11, arriving at $\mathrm{Vol}_d(P) \equiv_p 0$, since $\mathrm{Vol}_d(S_i) = 0$ for all $i$ as $d > 1 \geq \dim(S_i)$.

In the inductive step, assume $k \geq 2$ and that the assertion has been verified for $\mathrm{SU}^{k-1}(\mathcal{P}_0(\mathbb{Z}^n))$. Recall that every $P \in \mathrm{SU}^k(\mathcal{P}_0(\mathbb{Z}^n))$ can be written as $P = \sum_{i=1}^m \mathrm{conv}(A_i \cup B_i)$ for some polytopes $A_i, B_i \in \mathrm{SU}^{k-1}(\mathcal{P}_0(\mathbb{Z}^n))$. By the induction hypothesis, the $p^{k-1}$-dimensional normalized volumes of the $p^{k-1}$-dimensional faces of $A_i$ and $B_i$ are divisible by $p$. Consequently, by Proposition 13, the $p^k$-dimensional normalized volumes of the $p^k$-dimensional faces of $\mathrm{conv}(A_i \cup B_i)$ are divisible by $p$. Since $\mathrm{SU}^k(\mathcal{P}_0(\mathbb{Z}^n))$ is closed under taking faces (see Lemma 10), Proposition 11 applied to the $p^k$-dimensional faces of $P$ implies that the $p^k$-dimensional normalized volume of the $p^k$-dimensional faces of $P$ is divisible by $p$. $\qquad\square$

**Theorem 15.** *Let $d = p^t \leq n$ be a power of a prime number $p$, with $t \in \mathbb{N}$. Let $P$ be a lattice polytope in $\mathcal{P}_d(\mathbb{R}^n)$. If $h_P \in \mathrm{ReLU}_n^{\mathbb{Z}}(k)$, $k \in [t]$, then $\mathrm{Vol}_d(P)$ is divisible by $p$.*

*Proof.* By Corollary 9, we have $P + A = B$ for some $A, B \in \mathrm{SU}^k(\mathcal{P}_0(\mathbb{Z}^n))$. Then, by Proposition 11, one has $\mathrm{Vol}_d(P + A) \equiv_p \mathrm{Vol}_d(P) + \mathrm{Vol}_d(A) \equiv_p \mathrm{Vol}_d(B)$, which means that $\mathrm{Vol}_d(P) \equiv_p \mathrm{Vol}_d(A) - \mathrm{Vol}_d(B)$. By Theorem 14, both $\mathrm{Vol}_d(A)$ and $\mathrm{Vol}_d(B)$ are divisible by $p$. This shows that $\mathrm{Vol}_d(P)$ is divisible by $p$. $\qquad\square$

## 3.3 PROOFS OF MAIN RESULTS

We now turn to the proofs of Theorems 2 and 4. Let $N \in \mathbb{N}$ and recall that a rational number is an $N$-ary fraction if it is of the form $\frac{z}{N^t}$ with $z \in \mathbb{Z}$ and $t \in \mathbb{Z}_+$. For $N = 2$ and $N = 10$, one has binary and decimal fractions, used in practice in floating point calculations. Clearly, every binary fraction is also a decimal fraction, because $\frac{z}{2^t} = \frac{5^t z}{10^t}$.

In order to relate ReLU networks with fractional weights to ReLU networks with integer weights, we can simply clear denominators, as made precise in the following lemma.

**Lemma 16.** *Let $f : \mathbb{R}^n \to \mathbb{R}$ be exactly representable by a ReLU network with $k$ hidden layers and with rational weights all having $M$ as common denominator. Then $M^{k+1} f \in \mathrm{ReLU}_n^{\mathbb{Z}}(k)$.*

*Proof.* We proceed by induction on $k$. For the base case $k = 0$, $f$ is an affine function $f(x_1, \ldots, x_n) = b + a_1 x_1 + \cdots + a_n x_n$ with $b \in \mathbb{R}$ and $M a_1, \ldots, M a_n \in \mathbb{Z}$, from which the claim is immediately evident. Now let $k \geq 1$ and consider a $k$-layer network with rational weights with common denominator $M$ representing $f$. Then $f$ is of the form $f(x) = u_0 + u_1 \max\{0, g_1(x)\} + \cdots + u_m\{0, g_m(x)\}$ with $m \in \mathbb{N}$, where all $g_1, \ldots, g_m$ are functions representable with $k - 1$ hidden layers and all the weights, i.e., $u_1, \ldots, u_m$ and the ones used in

expressions for $g_1, \ldots, g_m$, are rational numbers with $M$ as a common denominator. Multiplying with $M^{k+1}$ we obtain

$$M^{k+1} f(x) = M^{k+1} u_0 + M u_1 \cdot \max\{0, M^k g_1(x)\} + \ldots + M u_m \cdot \max\{0, M^k g_m(x)\},$$

where the weights $M u_1, \ldots, M u_m$ are integer. By the induction hypothesis, for every $i \in [m]$, we have $M^k g_i \in \mathrm{ReLU}_n^{\mathbb{Z}}(k-1)$, and hence $M^{k+1} f \in \mathrm{ReLU}_n^{\mathbb{Z}}(k)$. $\qquad\square$

We are now ready to prove our main results.

*Proof of Theorem 2.* Let $k = \lceil \log_p(n+1) \rceil - 1$, i.e., $k$ is the unique non-negative integer satisfying $p^k < n+1 \le p^{k+1}$. If $F_n$ was representable by a ReLU network with $k$ hidden layers and $N$-ary fractions as weights, $\max\{0, x_1, \ldots, x_{p^k}\} = F_n(x_1, \ldots, x_{p^k}, 0, \ldots 0)$ would also be representable in this way. Thus, it suffices to consider the case $n = p^k$.

Recall that $F_n$ is the support function $h_{\Delta_n}$ of the standard simplex. Suppose, for the sake of contradiction, that $F_n$ can be represented by a ReLU network with $k$ hidden layers and weights being $N$-ary fractions. Let $t \in \mathbb{N}$ be large enough such that all weights are representable as $\frac{z}{N^t}$ for some $z \in \mathbb{Z}$. We use Lemma 16 with $M = N^t$ to clear denominators. That is, $N^{t(k+1)} F_n$ is representable by an integer-weight ReLU network with $k$ hidden layers. Since $F_n$ is homogeneous, we can assume that the network is homogeneous, too (Hertrich et al., 2021, Proposition 2.3). Observe that $N^{t(k+1)} F_n$ is the support function of $N^{t(k+1)} \Delta_n$, whose normalized volume satisfies $\mathrm{Vol}_n(N^{t(k+1)} \Delta_n) \equiv_p N^{nt(k+1)} \mathrm{Vol}_n(\Delta_n) = N^{nt(k+1)} \cdot 1 \not\equiv_p 0$. Hence, $N^{t(k+1)} \Delta_n$ is a polytope in $\mathbb{R}^{p^k}$ whose normalized volume is not divisible by $p$, but whose support function is represented by an integer-weight ReLU network with $k$ hidden layers. This contradicts Theorem 15. Hence, $F_n$ is not representable by a ReLU network with $k$ hidden layers and weights being $N$-ary fractions. $\qquad\square$

If $N = 10$, we can use $p = 3$ in Theorem 2, so Corollary 3 is an immediate consequence. The bound $\lceil \log_3(n+1) \rceil$ in Corollary 3 is optimal up to a constant factor, as $F_n$ is representable through a network with integral weights and $\lceil \log_2(n+1) \rceil$ hidden layers (Arora et al., 2018). A major open question raised by Hertrich et al. (2021) is whether this kind of result can be extended to networks whose weights belong to a larger domain like the rational numbers or, ideally, the real numbers.

We can also show that the expressive power of ReLU networks with weights being decimal fractions grows gradually when the depth $k$ is increasing in the range from 1 to $\mathcal{O}(\log n)$.

**Corollary 17.** *For each $n \in \mathbb{N}$ and each integer $k \in \{1, \ldots, \lceil \log_3 n \rceil\}$, within $n$-variate functions that are described by ReLU networks with weights being decimal fractions, there are functions representable using $2k$ but not using $k$ hidden layers.*

*Proof.* Function $F_{3^k}$ is not representable through $k$ hidden layers and weights being decimal fractions. Since $3^k \le 2^{2k}$, $F_{3^k}$ is representable with $2k$ hidden layers (and integer weights). $\qquad\square$

By making use of Theorem 2, we now present the proof of Theorem 4.

*Proof of Theorem 4.* To make use of Theorem 2, we need to find a prime number $p$ that does not divide $N$. Let $p_i$ denote the $i$-th prime number, i.e., $p_1 = 2, p_2 = 3, p_3 = 5$ etc. Moreover, assume that the prime number decomposition of $N$ consists of $t$ distinct primes, i.e., $N = p_{i_1}^{m_1} \cdots p_{i_t}^{m_t}$ where $m_1, \ldots, m_t \in \mathbb{N}$ and $i_1 < \cdots < i_t$. Choose the minimal prime $p$ that is not contained in $\{p_{i_1}, \ldots, p_{i_t}\}$, that is, the minimal prime not dividing $N$. Since $\{p_1, \ldots, p_{t+1}\}$ has a prime not contained in $\{p_{i_1}, \ldots, p_{i_t}\}$, we see that $p \le p_{t+1}$.

To get a more concrete upper bound on $p$, we make use of the prime number theorem (Hardy & Wright, 2008, Ch. XXII), which is a fundamental result in number theory. The theorem states that $\lim_{i \to \infty} \frac{p_i}{i \ln i} = 1$. Consequently, $p \le p_{t+1} \le 2t \ln t$ when $t \ge T$, where $T \in \mathbb{N}$ is large enough. We first stick to the case $t \ge T$ and then handle the border case $t < T$.

For $\ln N$ we have

$$\ln N = \sum_{j=1}^{t} m_j \ln p_{i_j} \ge \sum_{j=1}^{t} \ln p_{i_j} \ge \sum_{j=1}^{t} \ln(j+1) \ge \int_1^{t+1} \ln x \, \mathrm{d}x = (t+1)\ln(t+1) - t$$

for all $t \geq T$. Thus, $\ln N \geq \frac{1}{2} t \ln t$. This implies $\ln \ln N \geq \ln t + \ln \ln t - \ln 2$. Compare this to $\ln p \leq \ln 2 + \ln t + \ln \ln t$. So, we see that $\ln \ln N \geq C \ln p$ with an absolute constant $C > 0$. Hence, we can invoke Theorem 2 for $p$, getting that the number of layers needed to represent $F_n$ with integer weights is at least $\log_p n$, where $\log_p n \geq \frac{\ln n}{\ln p} \geq C \cdot \frac{\ln n}{\ln \ln N}$. In the case $t < T$, we observe that $p \leq p_T$ and obtain the lower bound $\log_p n = \frac{\ln n}{\ln p} \geq \frac{\ln n}{\ln p_T}$. Since $T$ is fixed, the factor $\ln p_T$ in the denominator is an absolute constant. $\qquad\square$

## 4 CONCLUSIONS

In summary, we proved that a lower bound for the number of hidden layers needed to exactly represent the function $\max\{0, x_1, \ldots, x_n\}$ with a ReLU network with weights being $N$-ary fractions is $\lceil \log_p(n+1) \rceil$, where $p$ is a prime number that does not divide $N$. For $p = 3$, this covers the cases of binary fractions as well as decimal fractions, two of the most common practical settings. Moreover, it shows that the expressive power of ReLU networks grows for every $N$ up to $\mathcal{O}(\log n)$. In the case of rational weights that are $N$-ary fractions for any fixed $N$, even allowing arbitrarily large denominators and arbitrary width does not facilitate exact representations of low constant depth.

Theorem 4 can be viewed as a partial confirmation of Conjecture 1 for rational weights, as the term $\ln \ln N$ is growing extremely slowly in $N$. If one could replace $\ln \ln N$ by a constant, the conjecture would be confirmed for rational weights, up to a constant multiple. As already highlighted in Haase et al. (2023), confirmation of the conjecture would theoretically explain the significance of max-pooling in the context of ReLU networks: It seems that the expressive power of ReLU is not enough to model the maximum of a large number of input variables unless network architectures of high-enough depth are used. So, enhancing ReLU networks with max-pooling layers could be a way to reach higher expressive power with shallow networks.

Methodologically, algebraic invariants – such as the $d$-dimensional volume $\mathrm{Vol}_d$ modulo a prime number $p$ when $d$ is a power of $p$ – play a key role in showing lower bounds for the depth of neural networks. Our proof approach provides an algebraic template for a general "separation strategy" to tackle problems on separation by depth. In the ambient Abelian group $(G, +)$ of all possible functions that can be represented within a given model, one has a nested sequence of subgroups $G_0 \subseteq G_1 \subseteq G_2 \subseteq \cdots$, with $G_k$ consisting of functions representable by $k$ layers. To demonstrate that an inclusion $G_k \subseteq G_{k+1}$ is strict, one could look for an invariant that can distinguish $G_k$ from $G_{k+1}$ – this is a group homomorphism $\phi$ on $G$ that is zero on $G_k$ but not zero on some $f \in G_{k+1}$. Most likely, the invariant needs to be "global" in the sense that, if $\phi(f)$ is computed from the NN representation of $f$, then it would accumulate the contribution of all the nodes of the NN in one single value and would not keep track of the number of the nodes and, by this, disregard the widths of the layers. In the concrete case we handled in this contribution, the group $G$ we choose is $\mathrm{ReLU}^{\mathbb{Z},0}$, whereas the invariant that we employ has values in $\mathbb{Z}_p$ and is based on the computation of the volume of lattice polytopes. In the original setting of Conjecture 1, one has to deal with the nested chain of subspaces $G_k = \mathrm{ReLU}^{\mathbb{R},0}(k)$ of the the infinite-dimensional vector space $G = \mathrm{ReLU}^{\mathbb{R},0}$, which makes it natural to choose as an invariant a linear functional $\phi \colon G \to \mathbb{R}$. To make further progress, it is therefore worthwhile continuing to investigate the connection between ReLU networks and discrete polyhedral geometry, algebra, and number theory in order to settle Conjecture 1 in general.

Finally, we raise a question on the role of the field of real numbers in Conjecture 1. Does the choice of a subfield of $\mathbb{R}$ matter in terms of the expressiveness? More formally, we phrase

**Question 18.** *Let $S$ be a subfield of $\mathbb{R}$ and $k \in \mathbb{N}$ and let $f \colon \mathbb{R}^n \to \mathbb{R}$ be a function expressible via a ReLU network with weights in $S$. If $f$ is expressible by a ReLU network with $k$ hidden layers and weights in $\mathbb{R}$, is it also expressible by a ReLU network with $k$ hidden layers and weights in $S$? What is the answer for $S = \mathbb{Q}$?*

If, for $S = \mathbb{Q}$, the answer to the above question is positive, then the version of Conjecture 1 with rational weights is equivalent to the original conjecture with real weights, which might be a helpful insight towards solving Conjecture 1. If the answer is negative, then the conjecture would have a subtle dependence on the underlying field of weights.

ACKNOWLEDGMENTS

We would like to thank Sergey Grosman from Siemens (Konstanz, Germany) for his feedback on the practical role of quantization. The research of GA was partially supported by the DFG Project 539867386 "Extremal bodies with respect to lattice functionals", which is carried out within the DFG Priority Program 2458 "Combinatorial Synergies".

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

## A   APPENDIX

### A.1   DEFERRED PROOFS

In this appendix, we provide the proofs missing in the main part of the article. For convenience of reading, we restate the corresponding statements.

#### A.1.1   PROOF OF LEMMA 10

This appendix provides the missing proof of the following lemma.

**Lemma 10.** *Let $k \in \mathbb{Z}_+$. Then, for all $P \in \mathrm{SU}^k(\mathcal{P}_0(\mathbb{Z}^n))$ and $u \in \mathbb{R}^n$, the face of $P$ in direction $u$, given by*

$$P^u \coloneqq \{x \in P \colon u^\top x = h_P(u)\},$$

*belongs to $\mathrm{SU}^k(\mathcal{P}_0(\mathbb{Z}^n))$. In other words, $\mathrm{SU}^k(\mathcal{P}_0(\mathbb{Z}^n))$ is closed under taking non-empty faces.*

*Proof.* Throughout the proof, let $\mathcal{X} = \mathcal{P}_0(\mathbb{Z}^n)$. The proof is by induction on $k$. For $k = 0$, we have $\mathrm{SU}^0(\mathcal{X}) = \mathcal{X}$. Since every polytope in $\mathcal{P}_0(\mathbb{Z}^n)$ consists of a single point $s$, every non-empty face of such a polytope also just consists of $s$, and is therefore contained in $\mathcal{P}_0(\mathbb{Z}^n)$. Thus, the claim holds.

Now let $k \geq 1$ and assume the assertion holds for $k - 1$. Furthermore, let $u \in \mathbb{R}^n$ and $P \in \mathrm{SU}^k(\mathcal{X})$ with $P = \sum_{i=1}^m \mathrm{conv}(A_i \cup B_i)$ for some $m \in \mathbb{N}$, $A_i, B_i \in \mathrm{SU}^{k-1}(\mathcal{X}), i \in [m]$. By definition and Minkowski additivity of the support function, we have $P^u = (\sum_{i=1}^m \mathrm{conv}(A_i \cup B_i))^u = \sum_{i=1}^m (\mathrm{conv}(A_i \cup B_i))^u$. Moreover, for each $i \in [m]$, $\mathrm{conv}(A_i \cup B_i)^u$ is equal to $A_i^u$, $B_i^u$, or $\mathrm{conv}(A_i^u \cup B_i^u)$ depending on whether $h_{A_i}(u) > h_{B_i}(u)$, $h_{A_i}(u) < h_{B_i}(u)$, or $h_{A_i}(u) = h_{B_i}(u)$, respectively. In any case, we obtain a representation of $P^u$ that shows its membership in $\mathrm{SU}^k(\mathcal{X})$, since $A_i, B_i \in \mathrm{SU}^{k-1}(\mathcal{X})$ for all $i \in [m]$ by the induction hypothesis. □

### A.1.2 PROOF OF PROPOSITION 13

The goal of this section is to prove the following statement.

**Proposition 13.** *Let $d = p^t \leq n$ be a power of a prime number p, with $t \in \mathbb{N}$. Moreover, let $P = \mathrm{conv}(A \cup B) \in \mathcal{P}_d(\mathbb{Z}^n)$ for $A, B \in \mathcal{P}_d(\mathbb{Z}^n)$. If $\mathrm{Vol}_{p^{t-1}}(F) \equiv_p 0$ for all $p^{t-1}$-dimensional faces F of A and B, then $\mathrm{Vol}_{p^t}(P) \equiv_p 0$.*

To prove this result, we need two auxiliary results that we provide next.

**Proposition 19.** *Let $m, s, d \in \mathbb{N}$ and $s < d \leq n$. If $P \in \mathcal{P}_d(\mathbb{Z}^n)$ such that $\mathrm{Vol}_s(F) \equiv_m 0$ for all s-dimensional faces F of P, then $\mathrm{Vol}_d(P) \equiv_m 0$.*

*Proof.* Note that we can restrict our attention to the case $d = s + 1$: Once the case $d = s + 1$ is settled, it follows that the divisibility of $\mathrm{Vol}_i(F)$ by $m$ for $i$-dimensional faces $F$ of $P$ implies divisibility of $\mathrm{Vol}_{i+1}(G)$ by $m$ for all $(i+1)$-dimensional faces $G$ of $P$. Hence, iterating from $i = s$ to $i = d - 1$, we obtain the desired assertion. So, assume $d = s + 1$.

Let $P$ be a $d$-dimensional lattice polytope with facets having a normalized $(d-1)$-dimensional volume divisible by $m$. We pick a vertex $a$ of $P$ and subdivide $P$ into the union of the non-overlapping pyramids of the form $\mathrm{conv}(\{a\} \cup F)$, where $F$ is a facet of $P$. By Proposition 7, the normalized $d$-dimensional volume of $\mathrm{conv}(\{a\} \cup F)$ is divisible by $\mathrm{Vol}_{d-1}(F)$. Since by assumption $\mathrm{Vol}_{d-1}(F)$ is divisible by $m$, we conclude that also $\mathrm{Vol}_d(P)$ is divisible by $m$, because $\mathrm{Vol}_d$ is additive as it is based on a Lebesgue measure. $\square$

The second result analyzes the structure of $\mathrm{conv}(A \cup B)$ in terms of a particular subdivision. Given a polytope $P \in \mathcal{P}(\mathbb{R}^n)$ of dimension $d$, a *subdivision* of $P$ is a finite collection $\mathcal{C} \subseteq \mathcal{P}(\mathbb{R}^n)$ such that (i) $P = \bigcup_{C \in \mathcal{C}} C$; (ii) for each $C \in \mathcal{C}$, the polytope $C$ has dimension $d$; (iii) for any two distinct $C, C' \in \mathcal{C}$, the polytope $C \cap C'$ is a proper face of both $C$ and $C'$. The elements $C \in \mathcal{C}$ are called the *cells* of subdivision $\mathcal{C}$, cf. (Lee & Santos, 2017).

**Proposition 20** (Haase et al. 2023, Prop. 10)**.** *For two polytopes $A, B \in \mathcal{P}(\mathbb{R}^n)$, there exists a subdivision of $\mathrm{conv}(A \cup B)$ such that each full-dimensional cell is of the form $\mathrm{conv}(F \cup G)$, where F and G are faces of A and B, respectively, such that $\dim(F) + \dim(G) + 1 = d$.*

The term "full-dimensional" in Proposition 20 as well as in the original formulation of Haase et al. (2023, Prop. 10) refers to faces that have the same dimension as $\mathrm{conv}(A \cup B)$, while its authors make no assumption on whether that dimension is equal to $n$ (but Haase et al. (2023) note in their proof that such an assumption would be without loss of generality).

We are now able to prove Proposition 13.

*Proof of Proposition 13.* Let $P = \mathrm{conv}(A \cup B)$. We apply Proposition 20 for obtaining a subdivision of $P$ into $d$-dimensional polytopes $P_1 = \mathrm{conv}(F_1 \cup G_1), \ldots, P_m = \mathrm{conv}(F_m \cup G_m)$, where for each $s \in [m]$, $F_s$ and $G_s$ are faces of $A$ and $B$, respectively, and $\dim(F_s) + \dim(G_s) + 1 = d$. That is, $P$ is the union of polytopes whose relative interiors are disjoint. Consequently, $\mathrm{Vol}_d(P) = \mathrm{Vol}_d(P_1) + \cdots + \mathrm{Vol}_d(P_m)$. It therefore suffices to show that $\mathrm{Vol}_d(P_s) \equiv_p 0$ for every such polytope $P_s$ with $s \in [m]$.

For given $s \in [m]$ and faces $F_s$ and $G_s$ of $A$ and $B$, respectively, denote their dimensions as $i$ resp. $j$. Since $i + j = d - 1 = p^t - 1$, the dimension of $F_s$ or $G_s$ is at least $p^{t-1}$ (if this was not the case, we would have $i + j \leq 2(p^{t-1} - 1) < p^t - 1$, which is a contradiction). By symmetry reasons, we assume without loss of generality that $i \geq p^{t-1}$. Then, by Proposition 19, $\mathrm{Vol}_i(F_s)$ is divisible by $p$. Consequently, by Proposition 7, the normalized volume of $\mathrm{conv}(F_s \cup G_s)$ is also divisible by $p$. $\square$

### A.2 PROOF OF BINOMIAL FORMULA FOR MIXED VOLUMES

The symmetry and multilinearity of the mixed-volume functional makes computations with it similar in nature to calculations with an $n$-term product. Say, the identity $(x+y)^2 = x^2 + 2xy + y^2$ over reals corresponds to the identity $\mathrm{Vol}_2(A+B) = \mathrm{V}(A+B, A+B) = \mathrm{V}(A, A) + 2\,\mathrm{V}(A, B) + \mathrm{V}(B, B) = \mathrm{Vol}_2(A) + 2\,\mathrm{V}(A, B) + \mathrm{Vol}_2(B)$ for planar polytopes $A, B$ and the way of deriving the latter identity

is completely analogous to deriving the identity for $(x + y)^2$ by expanding brackets. Very much in the same way, the binomial identity $(x + y)^n = \sum_{i=0}^{n} \binom{n}{i} x^i y^{n-i}$ corresponds to the identity (1). Here is a formal proof:

We use the notation $P_0 = B$ and $P_1 = A$. Then

$$\mathrm{Vol}_n(P_0 + P_1) = \mathrm{V}(P_0 + P_1, \ldots, P_0 + P_1)$$

by Property (c) in Theorem 5. Using Property (b) in Theorem 5 for each of the $n$ inputs of the mixed-volume functional, we obtain

$$\mathrm{Vol}_n(P_0 + P_1) = \sum_{i_1 \in \{0,1\}} \cdots \sum_{i_n \in \{0,1\}} \mathrm{V}(P_{i_1}, \cdots, P_{i_n}),$$

where the right-had side is a sum with $2^n$ terms. However, many of the terms are actually repeated, because $\mathrm{V}(P_{i_1}, \ldots, P_{i_n})$ does not depend on the order of the polytopes in the input: this mixed volume contains $i_1 + \cdots + i_n$ copies of $P_1$ and $n - (i_1 + \cdots + i_n)$ copies of $P_0$. Hence,

$$\mathrm{V}(P_{i_1}, \ldots, P_{i_n}) = \mathrm{V}(\underbrace{P_0, \ldots, P_0}_{n-(i_1+\cdots i_n)}, \underbrace{P_1, \ldots, P_1}_{i_1+\cdots+i_n}).$$

In order to convert our $2^n$-term sum into an $(n+1)$-term sum, for each choice of $i = i_1 + \cdots + i_n \in \{0, \ldots, n\}$, we can determine the number of choices of $i_1, \ldots, i_n \in \{0, 1\}$ that satisfy $i = i_1 + \cdots + i_n$. This corresponds to choosing an $i$-element subset $\{t \in [n]: i_t = 1\}$ in the $n$-element set $\{1, \ldots, n\}$. That is, the number of such choices is the binomial coefficient $\binom{n}{i}$. Hence, our representation with $2^n$ terms amounts to

$$\mathrm{Vol}_n(P_0 + P_1) = \sum_{i=0}^{n} \binom{n}{i} \mathrm{V}(\underbrace{P_0, \ldots, P_0}_{n-i}, \underbrace{P_1, \ldots, P_1}_{i}).$$

