# OpenReview forum: "On the Expressiveness of Rational ReLU Neural Networks With Bounded Depth"
_ICLR.cc/2025/Conference — ICLR 2025 Spotlight_

### Official Review · Reviewer_eki2 · 2024-10-20

**Soundness:** 2
**Presentation:** 4
**Contribution:** 3
**Rating:** 5
**Confidence:** 3

**Summary:**

This paper proves two lower bounds for the depth of a ReLU network with rational weights that exactly represents the maximum coordinate function.

**Strengths:**

1. This work extends the previous work by [Hasse et al., 2023]. It proves some statements for rational weighted neural networks, which is new to the literature. It seems to make a step towards understanding the exact functions representable by neural networks.

2. The presentation of the work is very clear. The results are very well-connected to the existing literature, their implications are made explicit, and the intuitions of the proofs are stated clearly.

**Weaknesses:**

1. The main weakness of this paper is the significance of the result. These analyses are interesting intellectual challenges, but I do not see too many practical implications. Studying the exact representations of neural networks is even a step back from the universal approximation results. This makes the results very brittle. For instance, I am not convinced by the argument that all weights in a computer are ultimately rational: while this is true, in a computer, one can neither represent the $F_n$ function studied in this paper exactly. Moreover, one can use a p-ary number to approximate any q-ary number arbitrarily well, which makes the results purely algebraic.

2. While the paper definitely contains interesting ideas, it seems to be fairly incremental and relies heavily on the mechanisms developed in [Hasse et al., 2023]. I don't see an exciting analysis tool that is invented in this work.

**Questions:**

1. Theorem 2 and 4 seem to have basically the same assumptions and yield two lower bounds. In which circumstances, should I care about one over the other?

2. Can you show or discuss the implications of Theorem 2 and 4 in the flavor of Conjecture 1 rather than approximation $F_n$?

---

> ### Author Response · Authors · 2024-11-18
> **Response to Review**
>
> Dear Reviewer eki2,
>
> thank you very much for your comments.
>
> Our contribution is classified as AI theory, and in this regard we hope that it holds theoretical significance in making progress towards resolving an interesting conjecture.
>
> However, we understand that numerical computations on a computer are ultimately only approximations of the exact operations on the rationals (or the reals), and the quality of approximation is among the key concerns in practice. Let us briefly give our take on why we think that exact representability (or rather: non-representability) results like ours are interesting for practitioners.
>
> Exact formulas are often applied in numerical and approximation contexts. Following this, possibility results (providing exact representations) are complemented naturally by impossibility results, which clarify when exact representations do not exist. Both aspects are significant and commonly studied together. It makes a difference whether there is an exact representation that one aims to approximate (say a limit of a sequence of solutions of slightly perturbed problems) or not. The former obviously makes error estimation much easier, while the latter can spell trouble in numerical practice.
> Indeed, https://arxiv.org/pdf/2203.01016 analyses the downsides when approximating the max function with fewer than logarithmic layers, and concludes that computational complexity of such approximations will be troublesome.
>
> Next to these general arguments, there is an area of research within AI that is closely related to rational-weight NNs: In order to save space needed to store a NN and boost the calculation speed when evaluating a given input, techniques for compressing the weights of the NN, known under the name *quantization*, have been deployed in AI and are currently widely used in practice (e.g., within the PyTorch package https://pytorch.org/tutorials/recipes/quantization.html), see https://arxiv.org/pdf/2103.13630 for a survey on quantization. In the case of so-called *uniform quantization* (which is most common) weights are essentially restricted to rationals with a fixed denominator, the point being that floating point arithmetic can be replaced with much faster INT8-operations (for example).
> We added a short reference on quantization to the paper as additional motivation. With our contribution we help to understand the impact of depth on the expressive power of quantized networks.
>
> If you feel the above arguments address you concerns, we would be glad if you consider raising your score. Moreover, any suggestions on how to better motivate our approach in the manuscript are of course appreciated.
>
> Concerning the mentioned weaknesses:
>
> While we build on [Haase et al, 2023], we aim to make interesting and non-trivial improvements. Extending from arguments modulo 2 to modulo any prime $p$ may seem straightforward in hindsight but actually involves nuanced insights.
> Furthermore, we apply the theory of mixed volumes, which generalizes and systematizes a key argument. This also suggests that mixed volume theory could be instrumental in fully resolving Conjecture 1. Indeed,  [Haase et al, 2023] raised several open problems, some of which could be resolved by our new techniques. For instance, the case of $2 \cdot F_n$ could not be investigated by [Haase et al, 2023], whereas we obtain a logarithmic bound for $2 \cdot F_n$, which is optimal up to a constant factor. Moreover, we can handle classes of rational numbers, for which the results of [Haase et al, 2023] were not applicable.
>
> Concerning the specific questions raised:
>
> - Theorem 1 serves as a starting point, highlighting the importance of prime factors of $N$ in our way of establishing lower bounds. Theorem 4 follows from this and its statement avoids number-theoretic specifics, instead addressing order-of-magnitude considerations in terms of $n$ and $N$.
>
> - Conjecture 1 claims that there is a gain in expressivity all the way up to $\log n$ layers, where saturation occurs. In the statement of this conjecture, it is easy to overlook its dependence on the weight domain. While the conjecture is stated over real numbers, other choices are possible, such as the field of rationals, real algebraic numbers, the ring of integers [Haase et al, 2023], or the ring of $N$-ary fractions, as used in our work. In all cases it has been shown previously that $F_n$ is one of the functions that are the hardest to represent exactly: If we can represent $F_n$, we can represent any function that is at all representable by a ReLU network. Thus one may replace $F_n$ by ``all functions representable by a ReLU network" in Theorems 2 and 4 (though we find the current formulation clearer). This impact of our results on expressiveness of neural networks is summarized in Corollary 17.

---

> > ### Comment · Reviewer_eki2 · 2024-11-21
> > **Thank you for your reply + Follow-up questions**
> >
> > Dear author(s),
> >
> > I appreciate your careful reply and I like your discussion of why people care about exact representations. My main lingering concern is that in your paper, you studied arbitrary $N$-ary rational weights. However, in all the useful scenarios that I could imagine, we need to fix an upper bound of the size of the denominator. This is the case, for example, in the quantization literature. I think the fact that you allow the denominator to grow arbitrarily large makes the results very brittle, and that is why I don't see too much practical gain from the original result on real-valued weights. I would like to invite the authors to comment on this. Thank you in advance.
> >
> > Reviewer eki2

---

> ### Author Response · Authors · 2024-11-23
> **Response to the Follow-up questions**
>
> Dear Reviewer eki2,
>
> thank you for your reply. We appreciate that you take the time to further discuss your concerns with us, and we are happy to provide additional comments on your follow-up question.
>
> We agree that in practice one would not work with arbitrary $N$-ary fractions, but impose a bound on the denominator. However, we would like to stress that if, in the case of bounded denominators, a common denominator $N$ of the weights is established, then all the weights are $N$-ary fractions with one place, as $1/N$ is written as $0{.}1$ in the $N$-ary system (see lines 77-78).
>
> In our paper, $N$ is arbitrary, but neither the base $N$ nor the actual denominators $N^t$ for some $t \in \mathbb{N}$ need to grow to infinity for any of our main results to hold (so neither $N$ nor $t$ need to be large). Although Theorem 4 involves a constant $C$ that is not further specified (which makes the theorem look asymptotic in nature), it still yields meaningful bounds for small values of $N$, while a value for $C$ could be traced down through the proof of Theorem 4. Note, however, that Theorem 4 involves estimates, so Theorem 2 will give a stronger bound in general.
>
> To give a concrete example, consider a network that includes max-pooling applied to, say, a 20x20 window of an image. Suppose we want to know how many layers would be needed to express such an operation ($F_{400}$) in a classical ReLU-network with weight denominators $N = 10$ (i.e., using one-place decimal fractions) if we did not explicitly use a max-pooling layer. Clearing denominators leads to the problem of modeling $10^{k+1} F_{400}$ by an integer-weight ReLU-network with $k$ hidden layers. For this case, results of [Haase et al. (2023)] do not provide any non-trivial bound on the depth. ([Haase et al. (2023)] explicitly mention that their results on $F_{n}$ do not exclude a sub-logarithmic integer-weight representation even for $2 \cdot F_{n}$.) From Theorem 2, however, we obtain the lower bound $\left \lceil{\log_3(400)}\right \rceil = 6$ for the number of hidden layers; if Conjecture 1 was proven, we could update that number to 9.
>
> Of course, it would also be interesting to investigate representability if we restrict the *numerator* as well. Since we wanted to stay close to Conjecture 1, we did not investigate this further and think it would go beyond the scope of this article.

---

> > ### Comment · Reviewer_eki2 · 2024-11-26
> >
> > Thank you for the further discussion. When I asked the follow-up questions, I failed to recall that the theorems proved are for lower bounds instead of upper bounds: it has been a while since I reviewed the paper and I am sorry for that. In that case, I agree that moving from the integer case to the rational case is a step forward. I have raised my score to the borderline, taking all discussions into account. My new evaluation was affected by both the contribution of the paper and my evaluation of the significance of the problem itself.

---

### Official Review · Reviewer_ojAS · 2024-10-28

**Soundness:** 4
**Presentation:** 4
**Contribution:** 3
**Rating:** 8
**Confidence:** 4

**Summary:**

The main result in the paper is a lower bound on the depth of the network for exactly representing the max function over the coordinates of the inputs. Namely, a ReLU feed-forward network with N-ary weights requires depth log_p(n+1) to represent the max over n coordinates, where p doesn’t divide N. Another result provides depth lower bound which relies on loglog(N).

**Strengths:**

1) The paper makes an important step toward solving the conjecture from Hertrich et al (2021), improving the previous depth lower bound which was restricted to networks with integer weights to networks with N-ary weights.

2) The presentation of the paper is clear and easy to follow.

3) I think the proof methods used in this paper are pretty unique in the literature of machine learning and may be used beyond the scope of this paper (although this could also be said to Haase et al. 2023).

**Weaknesses:**

I think this is a strong submission, and the weaknesses I write here are somewhat minor.

1) There are some definitions missing in the paper that are not common in the machine learning community, and it would be helpful to define everything formally. For example, semigroup (line 206). How is the dimension of a polytope defined (line 182)?

2) Although the paper is fairly self-contained, there are parts that are left without proof. Some examples: Lines 230-231. Why is SU closed under Minkowski addition? I think these are “easy” exercises but for a machine learning paper, it would be better to show it directly.

3) I suggest also citing https://arxiv.org/pdf/2006.00625, which proves that if a similar depth separation result for depth k>= 5 can be proven for approximation (rather than exact representation), then it would solve a longstanding open question in TCS about the separation of threshold circuits. In other words, it is highly unlikely that a similar result can be proven for approximation.

**Questions:**

1) Is there a more high-level intuition on why divisibility by p means that ReLU networks of a certain depth cannot approximate the target? The lower bounds on approximation usually use some measure of complexity that grows gradually with the depth, e.g. the number of linear regions that can be represented by the network. Is there a similar measure here?

2) Why the width doesn’t play any role in exact representation, contrary to approximation?

3) How do residual connections affect the result? Does the characterization of Theorem 8 still hold?

---

> ### Author Response · Authors · 2024-11-18
> **Response to Review**
>
> Dear Reviewer ojAS,
>
> thank you very much for your comments and your very positive assessment.
>
> Concerning the mentioned weaknesses:
>
> - We added brief definitions of the mentioned notions and added an argument why SU is closed under Minkowski addition, in the main part of the article where these concepts are introduced. The binomial formula for the mixed volumes is now proven in the appendix.
>
> - Thank you for pointing us to this very interesting reference. We do not expect that a depth-separation result similar to the one in our paper also holds for approximation, as [Safran et al., 2024] recently showed that $\max \{0,x_1,\ldots,x_n\}$ can be approximated arbitrarily well by $\text{ReLU}^{\mathbb{Z}}_n(2)$-networks of width $n(n + 1)$ https://arxiv.org/abs/2307.09212. We would have to look at bit closer at the details in both papers to see whether this result just rules out a non-approximability result for depth $\ge 4$ and small weights.
> In any case, we added the paper you suggested to our literature overview ([Safran et al., 2024] was already included).
>
> Concerning the specific questions raised:
>
> - We do not see a similarly intuitive complexity measure here. To provide an intuition, it might be worth looking at the case $k=1$ of just one hidden layer. This case covers the support functions
> of zonotopes, which are sums of finitely many segments. But, zonotopes can be decomposed into hyperparallelepipeds, all having the normalized volume divisible by $n!$. So, we know that the volume of the zonotope is divisible by $n!$. If $n$ is a prime power $p^t$, then $n!$ is divisible by $p$. So, the volume of the zonotopes is divisible by $p$, no matter how complex the zonotope is.
>
> - Our modular invariant does not directly capture the width-complexity of the underlying network, but rather it accumulates the contribution of the neurons in one single quantity (an element of $\mathbb{Z}_p$), from which the number of neurons cannot be read off directly but which still allows to carry out separation by the depth.
>
> - Allowing residual connections will not affect the result, since when we only consider expressiveness and there is no limit on the width, such connections can be simulated with classical ReLU networks.

---

> > ### Comment · Reviewer_ojAS · 2024-11-21
> >
> > I thank the authors for their detailed response and for addressing my concerns. I still think this is a strong submission that should be accepted, I will keep my score.

---

### Official Review · Reviewer_oyU3 · 2024-11-01

**Soundness:** 4
**Presentation:** 4
**Contribution:** 3
**Rating:** 8
**Confidence:** 4

**Summary:**

The authors provide new lower bounds on the number of layers that a ReLU neural network needs in order to represent the function $\max$ {$0, x_1, \ldots, x_n$}.
The main result is that if all weights are $N$-ary fractions and $p$ is a prime number that does not divide $N$, then at least $\lceil \log_p(n+1) \rceil$ layers are needed to represent the function $\max$ {$0, x_1, \ldots, x_n$}.
The proof relies on a duality between continuous piecewise linear functions and polytopes, as well as on the additivity of the mapping from lattice polytopes to their normalized volume modulo a prime number $p$. The authors extend results from Haase et al., who (implicitly) showed a lower bound of $\lceil \log_2(n+1) \rceil$ if the weights are $N$-ary fractions for odd $N$.
Additionally, for a given $N$, they provide an upper bound on a prime number $p$ that does not divide $N$, resulting in a non-constant lower bound for representing $\max$ {$0, x_1, \ldots, x_n$} with a rational neural network using $N$-ary weights.

**Strengths:**

The paper should be accepted because it strengthens results in an important research area. Additionally, the paper is very well-written.
- The expressiveness of ReLU neural networks is an active research field, and understanding the functions that are exactly representable with a specific architecture is a fundamental question. Finding lower bounds on the number of layers needed to compute all piecewise linear functions has remained an open problem for some years.
-  The lower bound of $\lceil \log_3(n+1) \rceil$ layers to represent the function $\max$ {$0, x_1, \ldots, x_n$} for neural networks with decimal or binary fractions is a notable result. It contributes further to resolving the conjecture of a non-constant lower bound on the number of layers needed to represent all piecewise linear functions.
-  The paper is well-organized, self-contained, and accessible. It effectively balances providing intuition with rigorous proofs.

**Weaknesses:**

While the results provide a meaningful strengthening, the methodology may appear somewhat incremental. The approach builds on Haase et al.'s method for networks with integer weights. Proposition 11 (the additivity of the mapping from a lattice polytope to its normalized volume modulo a prime number) is the main new technical component, derived from mixed volumes, which generalizes the parity of the normalized volume of certain polytopes associated with neural networks with integer weights to the divisibility of the normalized volume by a prime number.

That said, I still believe the paper is a valuable contribution, potentially opening the door to find other (algebraic) invariants for resolving (other special cases of) the conjecture. Moreover, the authors did an excellent job of bringing together these elements in a self-contained manner.

**Questions:**

- Do you see any potential to bound the denominator of the weights needed to represent the max function with fewer than logarithmic layers (and rational weights) so that your results would yield a non-constant lower bound?

Minor issues:
- On Line 151, should it be "faces of dimension at least $2^k$ for $k$ hidden layers"?
- The sentence in the conclusion, "In the case of rational weights, even allowing arbitrarily large denominators for any given $N$ and arbitrary width does not facilitate exact representations of low constant depth," might be slightly misleading. Perhaps consider phrasing it as "In the case of rational weights that are $N$-ary fractions..."

---

> ### Author Response · Authors · 2024-11-18
> **Response to Review**
>
> Dear Reviewer oyU3,
>
> thank you very much for your comments and your very positive feedback.
>
> We fully agree with your assessment that our proof relies on the strategy of [Haase et al, 2023], while the main new ingredient is using the theory of mixed volumes for proving Proposition 11. We are happy to read that you share our hopes that this idea might inspire further results that eventually lead to the resolution of the conjecture for more general weight domains.
>
> Regarding your question, we are unsure whether we understand it correctly. If you are asking for a lower bounds on the denominator of the weights needed to represent the max function with a given (small) number $k$ of layers, such a bound can be obtained from Theorem 4, by solving for $N$.
> Or are you asking for a result that would say if the max function can be represented with rational weights then it can also be represented with weights using denominators $\leq$ a certain constant and the same number of layers (i.e., is there an upper bound on the denominators that we may assume without loss of generality)? This would then indeed imply a non-constant lower bound on the depth for arbitrary rational weights. This looks like an interesting line of attack for Conjecture 1, but some new ideas would have to be generated to follow this path. Our results could be seen as a step in this direction, showing that allowing larger denominators can potentially only increase expressiveness if this also increases the number of primes dividing the denominator.
> Does that answer your question?
>
> Regarding the minor issues, we have corrected the statement (formerly) in line 151 and rephrased the sentence in the conclusion as suggested. Thank you.

---

### Official Review · Reviewer_1gfZ · 2024-11-03

**Soundness:** 3
**Presentation:** 3
**Contribution:** 3
**Rating:** 8
**Confidence:** 4

**Summary:**

In this paper, the authors identified the lower bound of the depth to exactly represent the function $F_n = \max (0,x_1,\dots,x_n) $. In particular, they focused on the rational neural networks whose weights are $N$-ary fractions, and showed that the depth $\lceil \log_p(n+1) \rceil$ and $\Omega(\frac{\ln n }{\ln\ln N})$ is necessary. These main results partially answer the conjecture by Hertrich et al. (2021).

**Strengths:**

**Extend previous works**

Independent of studying whether NNs can approximate a class of functions, identifying the class of functions that can be exactly implemented by NNs is one of the fundamental studies for the expressive power of NNs. The authors consider networks whose weights are $N$-ary fractions which represent floating point/fixed point arithmetic networks. Importantly, they exploit mixed-volume theory to prove the statement, which is not directly used in the previous works.
Overall, the results certainly help us have a better understanding of deep neural networks in view of a theoretical and practical way.

**Clarity and Quality**

well-written and clearly structured paper. The problem has been motivated nicely in the introduction. The related work section is rigorous and clearly details the current results for this problem.

**Weaknesses:**

**Partially answer conjecture 1 by Hertrich et al. (2021)**

Nonetheless, it is not a critical issue. As mentioned earlier, I think considering weights with $N$-ary fractions is more reasonable compared to real weights, since computers use floating/fixed point arithmetic. Thus, to highlight the novelty, it would be better to pose another conjecture, not focusing on conjecture 1, about a ReLU network using $N$-ary fractions weights.

**Proof technique**

Though the authors apply mixed-volume theory, most of the key lemma are based on previous works [Hertrich et al. (2021); Haase et al. (2023)].
Compared to the previous works, the proof strategy seems similar. Thus, I think this makes the contribution of this paper incremental. To highlight the novelty, it would be better to clarify why mixed-volume theory is necessary to extend the argument of Haase et al. (2023) (divisor 2 -> prime $p$).

**Questions:**

**Width and complexity**

As the depth, understanding the expressive power with respect to the width and # of parameters is also important. How do you expect the effect of the width or the number of parameters in your context? Suppose that $f$ can be represented by a ReLU network $g$ with $k$ hidden layers, but not with $k-1$ hidden layers. How do you expect the width and # of parameters of $g$? Can we use similar approaches (modulo, mixed-volume, etc.)?


**ReLU to other piecewise linear activation**

Once we shift our attention from ReLU to other piecewise linear activation (e.g. Leaky-ReLU), can we get similar results? Unlike ReLU, conjecture 1 may not be equivalent to the statement that any ReLU network representing $\max \\{ 0,x_1,...,x_n \\}$ requires $\log_2(n)+1$ hidden layers.

---

> ### Author Response · Authors · 2024-11-18
> **Response to Review**
>
> Dear Reviewer 1gfZ,
>
> thank you very much for your comments. We are pleased to see that you support acceptance of our work.
>
> Concerning the mentioned weaknesses:
>
> - Conjecture 1 is the strongest version among the possible variants of the conjecture for subrings or subfields of $\mathbb{R}$ as the underlying domain of weights. So, if Conjecture 1 (which is about real-valued weights) is true, the version with rational weights would be settled as a consequence. We thus see Conjecture 1 as the main conjecture. Nevertheless, the role of the domain for the weights is somewhat hidden in its formulation, and understanding the role of the domain could be a way to resolving it and its rational-weight version. We therefore have added Question 18 to the conclusions section of the paper, which essentially asks if there is any subfield of $\mathbb{R}$ for which Conjecture 1 would have a different answer.
>
> - As correctly pointed out, our proof follows the same high-level strategy as in [Haase et al, 2023]. The main new ingredient in our proof to strengthen their result is to make use of the theory of mixed volumes. This allows us to reveal that the divisor 2 used by [Haase et al, 2023] can be generalized to a prime number, which in hindsight might seem reasonable although its derivation requires to find the right tool, namely mixed volume theory. We believe that the idea to apply mixed volumes was non-trivial and could pave the way for making further progress towards resolving Conjecture 1, in particular, because we have only used a tiny part of the theory of mixed volumes. Our approach also allows to streamline some steps in the proof strategy. We therefore believe that our contribution is more than just incremental.
>
> Concerning the specific questions raised:
>
> - If $f$ is expressible by $k$ layers but not $k-1$ layers, the width necessary to represent $f$ can be arbitrarily large. Take $f = h_P$ to be the support function of a $2^k$-dimensional polytope $P \subseteq \mathbb{R}^n$ with $t$ vertices. Then $f$ admits a representation with $k$ layers. Since $P$ has $t$ vertices, the space $\mathbb{R}^n$ is decomposed into $t$ regions on which $f$ is piecewise linear. Since literature contains width bounds $b(n,k,w)$ on the number of regions of linearity that depend on the input size $n$, the depth $k$ and the width $w$, we see that when $t$ is chosen to be large enough to exceed $b(n,k,w)$, the function $f$ won't be representable with width $w$.
> We are currently not aware of any approaches that would use $\mathbb{Z}_p$ or mixed volumes to lower-bound the width, but it seems likely there are obstructions from discrete geometry that would help to cope with the width as well. While not within the scope of the current paper, this could be an interesting question for future research.
>
> - As the leaky ReLU activation function can be represented as the difference of two ReLU activation functions (and vice versa for a fixed slope for negative values), the expressive power of the network will not change in this case. Thus any statements on the expressive power under arbitrary width will transfer from one setting to the other. Consequently, the role of $\max \{0,x_1,\ldots,x_n\}$ will also remain the same. A similar argument can be made for activation functions with more than two linear pieces.

---

> > ### Comment · Reviewer_1gfZ · 2024-11-22
> >
> > Thank you for the detailed response. I am happy with the work's quality and contribution and have updated my score accordingly.

---

### Official Review · Reviewer_eDpF · 2024-11-04

**Soundness:** 4
**Presentation:** 4
**Contribution:** 3
**Rating:** 8
**Confidence:** 4

**Summary:**

The authors present new results about the impact of depth on the capacity of ReLU Neural Networks to exactly compute $F= x \mapsto \max \{ 0, x_1, \cdots, x_n \} $. More precisely, they investigate possible lower-bounds on the depth needed to compute $F$. This problem has direct consequences on the capacity of such Neural Networks to compute all piecewise-linear functions $\mathbb{R}^{n} \rightarrow \mathbb{R}$. They show that if the weights of the ReLU network are N-ary fractions (i.e. fractions of the form $ \frac{z}{N^{t}}$ where $z, N > 0, t \geq 0$ are integers), then there is a general logarithmic lower-bound.

**Strengths:**

- The paper makes progress towards a complete answer to an open and (believed to be) difficult conjecture.

- The class of ReLU neural networks considered are practically relevant.

- The technical parts are well-presented for readers not fluent in discrete geometry.

**Weaknesses:**

- There is no major weakness to this paper in my opinion. The minor weakness is that the paper is heavily inspired by a previous paper in the approach and only limited technical ingredients are added.

**Questions:**

Do the authors believe that an entirely new approach should be adopted to obtain results in the rational case?

---

> ### Author Response · Authors · 2024-11-18
> **Response to Review**
>
> Dear Reviewer eDpF,
>
> thank you very much for your comments and your positive assessment of our work.
>
> Concerning the mentioned weaknesses:
>
> Indeed, our paper relies on [Haase et al, 2023]. However, by using the theory of mixed volumes we were able to strengthen their result. We believe that this new ingredient was non-trivial and could pave the way for making further progress towards resolving Conjecture 1. Our approach also allows to streamline some steps in the proof strategy.
>
> Concerning the specific question raised:
>
> If one fixes the domain to be a field, such as the field of rational numbers, an analogous approach would be to find a linear function that separates the vector subspace of functions representable by $k$ layers from the larger space of functions representable by $k+1$ layers. On this level of detail, the approach could be similar, since our contribution uses a homomorphism of Abelian groups, acting onto $\mathbb{Z}_p$, for the purpose of separation. We believe that such a ``separation-based'' approach, which we follow, could be helpful for both the rational and the (original) real case. Since we believe that the question raised by you is important, we elaborate more on this in our update of the conclusions section.
>
> In the rational case, combinatorial invariants like divisibility of normalized volumes might play a crucial role further on and we hope that the theory of mixed volumes (only a tiny part of which we use in our contribution) will be a useful tool. However, the specific invariant given in Proposition 11 may not be enough for proving a stronger result and one has to look deeper into the combinatorics of the corresponding polytopes.

---

> ### Comment · Reviewer_eDpF · 2024-11-23
>
> Thank your for your detailed answer.
> After second evaluation of the content of the article, I confirm my first impression that the only downside of this work is that it essentially adapts existing techniques (normalized  and mixed volumes), to only a slightly more general case, i.e. $N$-ary fraction weights.
>
> However, this article is well-written, transparent about the core contributions and makes steady progress towards a complete answer to a mathematically precise conjecture about the expressivity of Neural networks.
>
> Furthermore, I believe this is a good opportunity to value work at the intersection of discrete geometry, polyhedral combinatorics to study neural networks and machine learning in general.
>
> Therefore, I think the positive aspects of this article outweighs its downsides, and I recommend its acceptance. I have updated my score accordingly.

---

### Author Response · Authors · 2024-11-18
**Comments on the revised version**

We thank all the reviewers for their valuable comments. We provide detailed answers to all their comments and questions in separate replies to the reviewers. We've uploaded a revision of the manuscript and summarize the key changes below.

The detailed overview of the changes is provided in the diff attached as a supplementary material.

- Extended motivation for the $N$-ary fractions setting by mentioning quantization in the introduction.
- Conclusion section has been considerably extended by an overview of the separation approach and the role of the choice of the weight domain in Conjecture 1.
- A proof of the binomial formula for mixed volumes is added to the appendix for the sake of completeness.

We also slightly extended the literature overview, added definitions for basic algebraic notions and corrected several typos.

---

### Meta-Review · Area_Chair_ssev · 2024-12-05

**Metareview:**

A fine work that makes an elegant contribution to the theory of ReLU neural networks by providing depth lower bounds for representing the max function under rational weights. Its execution, rigor, and progress on a practically relevant topic make it a standout submission. The authors are encouraged to incorporate the important feedback given by the knowledgeable reviewers.

**Additional Comments On Reviewer Discussion:**

The overall evaluation improved during the discussion period.

---

### Decision · Program_Chairs · 2025-01-22

Accept (Spotlight)